# Effects of Caloric Intake and Aerobic Activity in Individuals with Prehypertension and Hypertension on Levels of Inflammatory, Adhesion and Prothrombotic Biomarkers—Secondary Analysis of a Randomized Controlled Trial

**DOI:** 10.3390/jcm9030655

**Published:** 2020-02-28

**Authors:** En-Young N. Wagner, Suzi Hong, Kathleen L. Wilson, Karen J. Calfas, Cheryl L. Rock, Laura S. Redwine, Roland von Känel, Paul J. Mills

**Affiliations:** 1Department of Psychiatry, University of California, San Diego, CA 92093, USA; s1hong@ucsd.edu (S.H.); k8wilson@ucsd.edu (K.L.W.); pmills@ucsd.edu (P.J.M.); 2Department of Consultation-Liaison Psychiatry and Psychosomatic Medicine, University Hospital Zurich, 8091 Zurich, Switzerland; roland.vonkaenel@usz.ch; 3Department of BioMedical Research, Inselspital, Bern University Hospital, 3010 Bern, Switzerland; 4Department of Family Medicine and Public Health, University of California, San Diego, CA 92093, USA; kcalfas@ucsd.edu (K.J.C.); clrock@ucsd.edu (C.L.R.); 5College of Nursing, University of South Florida, Tampa, FL 33612, USA; lredwine@health.usf.edu

**Keywords:** hypertension, prehypertension, exercise, diet, adhesion molecule, sICAM, soluble intercellular adhesion molecule, inflammatory markers, intervention

## Abstract

Background: Cardiopulmonary fitness and low calorie diets have been shown to reduce inflammation but few studies have been conducted in individuals with elevated blood pressure (BP) in a randomized intervention setting. Thereby, adhesion biomarkers, e.g., soluble intercellular adhesion molecule (sICAM)-3, have not been examined so far. Methods: Sixty-eight sedentary prehypertensive and mildly hypertensive individuals (mean age ± SEM: 45 ± 1 years; mean BP: 141/84 ± 1/1 mmHg) were randomized to one of three 12-week intervention groups: cardio training and caloric reduction, cardio training alone, or wait-list control group. Plasma levels of inflammatory, adhesion and prothrombotic biomarkers were assessed. In a second step, intervention groups were combined to one sample and multivariate regression analyses were applied in order to account for exercise and diet behavior changes. Results: There were no significant differences among the intervention groups. In the combined sample, greater caloric reduction was associated with a larger increase of sICAM-3 (*p* = 0.026) and decrease of C-reactive protein (*p* = 0.018) as a result of the interventions. More cardio training was associated with increases of sICAM-3 (*p* = 0.046) as well as interleukin-6 (*p* = 0.004) and a decrease of tumor necrosis factor-α (*p* = 0.017) levels. Higher BP predicted higher plasminogen activator inhibitor (PAI)-1 (*p* = 0.001), and greater fitness predicted lower PAI-1 levels (*p* = 0.006) after the intervention. Conclusions: In prehypertensive and hypertensive patients, plasma levels of the adhesion molecule sICAM-3 and inflammatory biomarkers have different response patterns to cardio training with and without caloric reduction. Such anti-inflammatory and anti-thrombotic effects may have implications for the prevention of atherothrombotic cardiovascular disease among individuals at increased risk.

## 1. Introduction

The pathogenesis of hypertension seems to be more complex, as it is difficult to treat despite established antihypertensive drugs and longtime cardiovascular research. Hypertension (blood pressure (BP) ≥ 140/90 mmHg) is a highly prevalent disease among United States (U.S.) citizens with a well-known impact on numerous comorbid conditions and public-health related outcomes [1,2,3]. On a cellular level, it is established knowledge that individuals with hypertension exhibit elevated blood levels of cellular adhesion molecules [4,5,6] as well as inflammatory markers [7,8,9,10,11,12,13,14,15]. The reason for that is mainly seen in endothelial activation [12], which explains the tight linkage between elevated BP, chronic inflammatory processes including atherosclerosis and adverse health related outcomes [7,16]. So far, studies on the effects of human hypertension on adhesion molecules have been focusing on soluble intercellular adhesion molecule-1 (sICAM-1; sCD54) [4,5,6] while studies measuring sICAM-3, which is potentially much more important for immune response [17], are scarce.

In treating hypertension, lifestyle modifications including adopting a regular exercise regimen are key constituents of treatment [18,19]. This recommendation might also apply to prehypertension (BP > 120/80 and <140/90 mmHg), a risk factor for the development of hypertension. The effect achieved depends on both intensity and frequency of exercise [20,21,22], and BP reductions of up to 15 mmHg can be achieved without using antihypertensive drugs and thus avoiding their common side effects [23]. In overweight hypertensive subjects, a combined exercise and dietary weight-loss intervention was shown to lead to even greater reductions in BP than exercise alone [22].

Meanwhile, few studies have examined the relationship between exercise and inflammation in hypertensive patients. One of the few studies showed that exercise and/or exercise plus diet interventions lead to reductions in circulating interleukin (IL)-6 and C-reactive protein (CRP) levels in obese individuals [24]. In normotensive, healthy individuals, cross-sectional epidemiological studies have shown physical activity to be negatively associated with CRP, fibrinogen, and white blood cell count [25] and inflammatory markers, including tumor necrosis factor (TNF)-α and IL-6 and prothrombotic plasminogen activator inhibitor (PAI-1) were shown to be inversely related to physical activity levels [26,27,28]. Furthermore, our group has shown that physical activity level and fitness are inversely related to leukocyte adhesion molecule expression [29,30].

Guided by the above described effects observed in healthy individuals, the purpose of this study was to explore the relationship of a 12-week exercise (and diet) intervention and inflammation/cell adhesion in patients with prehypertension and hypertension by measuring circulating levels of CRP, IL-6, TNF-α, PAI-1, and sICAM-1 and sICAM-3 by taking relevant cardio-metabolic confounding factors (body mass index (BMI), physical inactivity) into account. Based on these data of a randomized controlled trial (RCT), our aim was to show for the first time in vivo that exercise has not only an impact on levels of inflammatory biomarkers, but also on levels of cellular adhesion molecules. Our main hypothesis was that, compared to a 12-week wait-list control group, the 12-week exercise intervention, and especially the 12-week exercise plus diet intervention, would result in greater reduction of inflammatory biomarkers and cell adhesion molecules pre to post intervention. Second, we expected that both intervention groups would show increased fitness levels and greater BP reduction and weight reduction as compared to the wait-list control group.

## 2. Methods

### 2.1. Ethical Approval and Informed Consent

Data is taken from a clinical trial on the effects of exercise and diet on inflammation in hypertensive individuals registered in a public registry (www.clinicaltrials.gov, ClinicalTrials.gov Identifier: NCT00338572). The University of California San Diego (UCSD) Institutional Review Board approved the study protocol, to which all participants signed a written informed consent. All procedures performed in studies involving human participants were in accordance with the ethical standards of the institutional and/or national research committee and with the 1964 Helsinki Declaration and its later amendments or comparable ethical standards. Participants were recruited from the local community through flyers, referrals from local doctor’s offices and e-mail list serve distribution.

### 2.2. Data Source, Effect of Exercise and Diet on Inflammation in Hypertensive Individuals Trial

The data of the present secondary analysis is derived from a prospective, randomized controlled intervention study examining the effect of exercise and diet on inflammation in hypertensive individuals. That study was conducted in order to compare the effect of an exercise program versus a combined exercise and diet program on reducing inflammation in hypertensive individuals. Figure 1 shows the RCT CONSORT flow diagram for participants. Of 168 eligible people, 77 individuals chose not to enter the RCT. The 91 remaining participants were randomly assigned to one of three 12-week intervention groups: aerobic cardio training and caloric reduction, aerobic cardio training alone, or wait-list control group. Participants assigned to the wait-list group were controls for at least 12 weeks until they were allocated to one of the intervention groups (exercise alone or exercise plus “Dietary Approaches to Stop Hypertension” (DASH) group). Of these 91 subjects, 9 individuals did not complete the study for personal reasons, moving away from San Diego (*n* = 2), becoming too busy or starting a new job (*n* = 5), becoming ill (*n* = 2, colon rupture, neurological concerns), and 14 individuals were not reliable/cancelled.

The trial included relatively healthy, sedentary and non-medicated men and women (*n* = 91) between age 18 and 65 years with prehypertension (>120/80 but <140/90 mmHg) and stage 1 and 2 hypertension (≥140/90 but <180/110 mmHg), who were not on a current diet or had been participating in an exercise program within the past 6 months. The level of habitual physical activity was assessed using the Leisure Time Exercise Questionnaire (LTEQ) by Godin and Shepard [31] to confirm the sedentary status; individuals with LTEQ total scores above 40 were excluded. Major concomitant diseases were ruled out by blood sample, resting electrocardiogram and medical history. For each eligible participant, a trained technician or registered nurse obtained anthropometric data through standard procedures. Body weight, height and average resting BP (six measurements taken after a 15-min seated rest on two separate days using a Dinamap Compact BP^®^ monitor (Critikon, Tempa, FL, USA)). Participants with values of high-sensitivity CRP >10 mg/L were excluded from the analyses, as values above this threshold are indicative of an acute infection.

### 2.3. Fitness Testing

Participants underwent a maximum oxygen intake (VO_2peak_, mL/kg/min) treadmill exercise test to determine cardiorespiratory fitness using the standard Bruce protocol in which treadmill speed and grade were increased gradually from 1.7 mph and 10% grade every 3 min. Expired gas was analyzed by Sensormedics metabolic cart, Milano, Italy (Vmax software version 6-2A).

### 2.4. Aerobic Cardio Training

Since all participants were relatively sedentary at baseline, the training intervention was designed to increase exercise levels gradually to the ultimate physical activity goal: Five or more days per week of at least moderate-intensity cardiovascular physical activity (e.g., walking) for 30 to 60 min. Exercise intensity was individually adapted by the assessed heart rate (before and during exercise). Participants met with a certified personal trainer (study personnel) at the local YMCA 2 days a week for the entire 12-week intervention period. They shared with the trainer the fitness information. In-person sessions with a trainer were scheduled, as they are more likely to be completed than entirely home-based exercise and may increase the motivation for completion of the three sessions per week of exercise without a personal trainer. The sessions with the trainer included training in proper warm up, exercise and cool down and stretching. The participant was asked to do physical activity on his or her own (either in their home, neighborhood or at the YMCA) for three additional days per week. In addition to meeting a goal for the number of minutes of physical activity, participants were asked to increase incidental physical activity. To monitor their progress, participants were given a pedometer and asked to work up to an ultimate goal of an average of 10,000+ steps per day. We used the same pedometer (Omron^®^, Kyōto, Japan) throughout the study to warrant an adequate comparison. The inclusion of the pedometer and step goal helped to ensure that participants were not compensating for increased activity during their sessions with the personal trainer by being more sedentary throughout the rest of the day. The achievement of these goals ensured an increase in energy expenditure, which contributed to a negative energy balance. Subjects completed a log of their daily pedometer readings and presented them to the trainer each week.

### 2.5. Caloric Reduction

The dietary intervention was led by research dietitians at the Nutrition Services Core of the UCSD Medical Center General Clinical Research Center (GCRC) or Clinical and Translational Research Institute (CTRI). The core of the diet plan was based on the DASH clinical study [32,33]. This dietary pattern is high in fruits, vegetables, and low fat dairy foods, and low in saturated and total fat. It also is high in dietary fiber, potassium, calcium, and magnesium, and moderately high in protein. The dietary intervention aimed at reducing the individual energy intake by 500–1000 kcal/day taking into account lifestyle and dietary preferences of the participant [34]. In addition, concomitant behavioral strategies like conscious eating and stimulus control were trained. Thus, participants received intensive intervention approaches known to produce behavior change. Subjects in the experimental group met with registered dietitians and/or certified exercise trainers to establish initial dietary and physical activity goals. Regular meetings in person and by phone continued for the entire 12-week intervention. All subjects recorded their food (caloric intake in kilojoule) and provided the information to the study investigators. Diet adherence was assessed with three daily dietary recalls administered by the study dietician.

### 2.6. Assessment of Inflammatory Biomarkers and Cellular Adhesion Molecules

Fasting blood samples were collected within one week of, but at least two days after, the maximum exercise test at baseline. All assays were performed in the UCSD Clinical Research Biomarker Laboratory using commercially available immunoassay kits (R&D Systems products, Minneapolis, MN, or MSD Systems, Rockville, MD, USA). Intra- and inter-assay coefficients of variation (CV) were confirmed to be <5%. For subsequent statistical analyses, soluble ICAM-1 (sample size, 63), sICAM-3 (66), PAI-1 (68), CRP (62), IL-6 (65), and TNF-α (63) measures were log-transformed to normalize distributions with differing sample sizes reflecting missing biomarker data due to technical errors, etc., and we calculated the difference (Δ) from pre to post intervention levels for all biomarker measures by detracting the value before the intervention from the value after the intervention.

### 2.7. Secondary Analysis of Trial Data

According to the RCT protocol, *n* = 18 had been to the aerobic cardio training and caloric reduction intervention, *n* = 28 to the aerobic cardio training alone intervention and *n* = 22 to the wait-list control group. However, in reality, participants began modifying their exercise or diet behavior regardless of their intervention group assignment and modest group sizes limited statistical power. Therefore, we decided for a statistical approach with a secondary analysis of data from all 68 participants combined that may better warrant to indicate the actual behavior and inflammatory processes and consider the intervention effect described respectively.

Thus, a series of multivariate linear regression analyses were performed to examine the role of the different intervention elements (predictors). For the three predictors fitness, cardio training, and caloric intake, the differences (Δ) from pre to post intervention levels were used in the statistical analyses. We calculated these differences (Δ) by detracting the value before the intervention from the value after the intervention. Associations between the four predictors (BP, fitness, cardio training, and caloric intake) and outcomes (differences (Δ) from pre to post intervention levels of cellular adhesion molecules (sICAM-1, sICAM-3) and inflammatory biomarkers (CRP, IL-6, TNF-α)) were determined using multiple linear regression models, separately for each of the five outcome variables. Results were considered statistically significant at *p* ≤ 0.05. We did not adjust p-values for multiple comparisons, as the main primary outcome was CRP; however, as we examined the potential interaction among inflammatory, adhesion and prothrombic markers, we defined one primary outcome marker in each of these three biomarker domains playing different roles in CVD: CRP for inflammation, sICAM-1 for adhesion molecules, and PAI-1 for prothrombotic factors. All other outcomes (IL-6, TNF, sICAM-3) were secondary (Il-6 and TNF) or exploratory (sICAM-3, as no study so far has been investigating sICAM-3).

For the associations between each of the predictors and inflammatory marker outcome levels, two models of increasing complexity were computed: in Model 1 we adjusted for (prehypertensive or hypertensive) BP and sociodemographic factors (age, gender) only, and in Model 2 we additionally adjusted for the other remaining predictors in one complete regression model (Table 1). Only the significant results of the five statistical models are shown for a better overview (Table 1). For our second hypothesis, mean pre and post intervention BP, fitness levels, and weight of the three groups were compared to each other. All statistical analyses were performed using IBM^®^ SPSS^®^ version 22.0 statistical software package (IBM Corporation, New York, NY, USA).

## 3. Results

### 3.1. Study Sample

The baseline demographic, metabolic and health behavior characteristics of all study participants (*n* = 68) included in the statistical analyses are displayed in Table 2. In the combined 68 participants, 22 subjects had reduced caloric intake and exercised regularly. These 22 participants had an average daily caloric reduction of 673 ± 594 calories (mean ± SEM) and had on average exercised 22.9 ± 2.8 h during the 12-week intervention (on average 114.5 ± 14 min per week). The intensity of aerobic cardio training of the hypertensive patients did not differ significantly from that of the prehypertensive patients (*p* = 0.18). Subjects who exercised regularly but had no caloric reduction (*n* = 38) had spent on average 21.1 ± 2.0 h on training during the 12-week intervention (on average 105.5 ± 10 min per week). Thirteen % of the participants were current smokers based on self-report.

### 3.2. Blood Pressure Classification before and after Intervention

After the intervention the mean SBP reduction for hypertensive subjects (*n* = 34) with 10.1 ± 2.6 mmHg was greater than the prehypertensive subjects (*n* = 25) with 4.4 ± 1.3 mmHg but not significantly so (*p* = 0.06). However, the average SBP of the hypertensive patients (*n* = 34; mean ± SEM: 148.1 ± 1.6 mmHg) decreased to the prehypertensive level of 137.9 ± 2.3 mmHg.

### 3.3. Cellular Adhesion Molecules and Inflammatory Biomarker Associations with Elevated Blood Pressure, Fitness and Obesity in the 68 Participants

We calculated Pearson’s correlations to examine the associations of the soluble intercellular adhesion molecules and inflammatory biomarkers with BP, BMI, and fitness before the intervention. CRP was negatively correlated with fitness (*r* = −0.308; *p* < 0.05) and positively with BMI (*r* = 0.327; *p* < 0.01). IL-6 was negatively associated with fitness (*r* = −0.311; *p* < 0.05) and positively associated with SBP (*r* = 0.310; *p* < 0.05). TNF-α was negatively correlated with SBP (*r* = −0.364; *p* < 0.01). Soluble ICAM-1 was negatively associated with fitness (*r* = −0.242; *p* < 0.05) and was positively associated with CRP (*r* = 0.414; *p* < 0.01) but not with BP. Soluble ICAM-3 had no associations with BP, BMI, fitness or sICAM-1.

### 3.4. Behavioral Change by Intervention Elements, Change of BMI and Fitness and Its Role in Inflammation in All Participants with Elevated Blood Pressures

The higher the SBP the higher were PAI-1 levels (β = 0.427; *p* = 0.001), and the more the fitness level had improved during the intervention the lower were PAI-1 levels (β = −0.392; *p* < 0.01) after the intervention (Table 1). Aerobic cardio training was a significant independent predictor of sICAM-3, IL-6, and TNF-α levels after controlling for sociodemographic variables and SBP in a first step regression model (Table 1), meaning the greater the amount of total moderate exercise was, the higher were the levels of sICAM-3 and IL-6, and the lower were the levels of TNF-α (Table 1) after the intervention. From earlier analyses [35,36], BMI decreased in the exercise groups, and fitness increased in those groups. These results remained statistically significant, even after adjusting for changes in BMI, fitness, and caloric reduction in the final model (sICAM-3 (β = 0.301; *p* < 0.05), IL-6 (β = 0.415; *p* < 0.01), and TNF-α (β = −0.357; *p* < 0.05); Table 1). Caloric reduction (to reduce caloric intake) predicted significantly higher sICAM-3 (β = −0.332; *p* < 0.05) and lower CRP (β = 0.342; *p* < 0.05) levels in the first step and final model (Table 1).

## 4. Discussion

We examined the differential effects of aerobic cardio training and caloric reduction on inflammatory, adhesion and prothrombotic biomarkers in prehypertensive and hypertensive subjects. Our study adds to the existent body of literature in the context of behavioral (aerobic exercise and diet) intervention and hypertension for inflammatory biomarker outcomes. The more comprehensive approach of lifestyle changes including exercise and diet behavior change of this 12-week-intervention study might explain the differential changes of inflammatory (CRP, IL-6, TNF-α) and prothrombotic (PAI-1), and adhesion (sICAM-1 and -3) biomarkers.

The key finding is that aerobic cardio training and/or caloric reduction had differential effects on IL-6 and adhesion molecule sICAM-3 in vivo. In particular, the latter finding is novel showing a prospective association between exercise and caloric reduction respectively and sICAM-3 levels. Thereby, this is the first study to our knowledge that shows that aerobic exercise has not only an impact on inflammatory biomarkers but also on levels of cellular adhesion molecules. Until now, it was unclear how they might interact, although it is well known that aerobic exercise reduces inflammatory processes.

The increase of sICAM-3 and IL-6 in subjects with elevated BP was independently predicted by the intensity of aerobic cardio training and caloric reduction. We think that this finding may be initially counter-intuitive but is important for the following reasons: First, this finding is consistent with prior studies, which showed that exercise was associated with a significant increase in IL-6 [37,38]. In this study, however, increased IL-6 levels after intervention and its positive correlation with both exercise levels and caloric reduction may be explained by potentially anti-inflammatory effects of IL-6 [39]. IL-6 is a pleiotropic cytokine released by skeletal myocytes under contraction, mediating anti-inflammatory effects by stimulating the production of anti-inflammatory cytokines and suppression of TNF-α production [39] with all these mechanisms concurring with the results of this study. Second, this is the first study to highlight the potential role of sICAM-3 in vascular inflammatory effects of exercise and/or diet intervention among prehypertensive and hypertensive subjects. To date, studies on the effects of human hypertension on adhesion molecules have been focusing on sICAM-1 levels, which are increased in hypertensive individuals [4,5,6]. Soluble ICAM-1 levels reflect ICAM-1 expression on activated endothelial cells [40] and are associated with the degree of atherosclerosis [41]. Although ICAM-1 and ICAM-3 share a similar immunoglobulin-like structure and amino acid identity of about 48%, with the greatest homology observed in domains 2 and 3 [42], their differential pattern of expression and cellular distribution suggest a different functional role [42]. The interactions of ICAM-1 and ICAM-3 with lymphocyte function-associated antigen (LFA)-1 are integral to the normal functioning of the immune system [42]. ICAM-3 is expressed on resting leucocytes and is potentially the most important ligand for LFA-1 in the initiation of the immune response because the expression of ICAM-1 on resting leucocytes is low [17]. Although sICAM-3 binds to LFA-1 with an affinity approximately nine-times weaker than sICAM-1, sICAM-1 and sICAM-3 compete with each other for binding to LFA-1 [42]. As we did not find changes in sICAM-1 levels post interventions in this study, we speculate that the increase in sICAM-3 levels as a result of changes in exercise and/or diet behavior in this study may exhibit a “buffer” for the sICAM-1 function, potentially leading to down-regulating vascular endothelial inflammation. It is not fully understood whether the increase of sICAM-3 by cardio training and caloric reduction in prehypertensive and hypertensive subjects reflects a cardio-protective mechanism. Given the lack of knowledge in the vascular function of sICAM-3 and its implications as a vascular inflammatory biomarker, a follow-up investigation is necessary to examine its vascular action and interactions separate of or in conjunction with other better-known vascular endothelial markers such as sICAM-1.

Another relevant potential underlying pathway in hypertension, metabolic syndrome, dyslipidaemia, and abdominal obesity, is the dysregulation of the hypothalamic–pituitary–adrenal (HPA) axis leading to catecholamine activation of β-adrenergic receptors (β-ARs) [43]. This may lead to altered leukocyte/endothelial binding, endothelial injury with up-regulation of endothelial ICAM-1 expression and increase of inflammatory cytokines [44]. In this context, regular aerobic exercise has also been shown to enhance β-AR sensitivity in the context of age-related decline in adrenergic responsiveness, i.e., increase of total peripheral resistances, which is a relevant factor underlying hypertension, atherosclerosis, and vascular insufficiency [45,46].

This study has a number of limitations. First, a weakness of the experimental design of this study is the lack of healthy BP controls. Thus, we are unable to determine whether the inflammatory, adhesion, and prothrombotic factors measured were indeed elevated in our sample of individuals with hypertension and prehypertension as compared with healthy individuals, and whether the changes induced by diet and exercise in these factors are at the level of clinical benefits. Second, a randomized and controlled trial of 12-week intervention of health behavior change was proven challenging and resulted in modest group sizes, limiting statistical power. More importantly, few significant differences found among intervention groups led us to scrutinize the actual changes in exercise and diet behavior. In reality, in spite of our best efforts to avoid “cross-contamination” between health behaviors, participants naturally began modifying their exercise or diet behavior regardless of their intervention group assignment. Some participants assigned to wait-list control condition also began to change their health behavior. Although troubling for behavior intervention trials, this kind of behavior is not particularly surprising, as exercise and healthy diet behaviors tend to co-occur [47]. It was revealed also particularly challenging to prohibit study volunteers who exhibit “readiness” to change their health behavior from doing so to follow randomization to a wait-list control group in a trial such as this. Therefore, as aforementioned, we collapsed all groups into one sample and applied multivariate regression analyses to the whole sample in order to account for exercise and diet behavior changes in practice during the intervention period. Third, regarding the dietary intervention, we have to mention that we did not differ short supply chain (SSC) from long supply chain (LSC) food included in the DASH [48]. A recent cross-sectional study found that only individuals adhering to the Mediterranean diet with an SSC had a significantly reduced prevalence of metabolic syndrome compared to those with an LSC [48]. Further, recent studies have shown that inflammatory effects of food parameters exist with a high variability [49] due to either a different genetic pattern or to diverse genetic–environmental interactions, and potential interactions (synergism, antagonism) among food parameters. However, as previously proposed, such a “dietary inflammatory index” ought to be advanced further [49]. Another intervention group, diet alone, may have contributed to important additional information. Fourth, our study did not explore the role of gender in the potentially differing effects of exercise and/or diet on inflammatory or prothrombotic markers, which was beside the primary aim of the study and limited by a limited sample size. Future studies should investigate the effect of gender not only on the intervention effects but also such behavior changes.

In summary, regular exercise and caloric reduction in individuals at risk of essential hypertension may have differential effects on biomarkers of CVD risk which may have clinical implications for the prevention of atherothrombotic cardiovascular disease. Assessments of the actual behavioral changes above and beyond intervention group assignment appear to be important. Nonetheless, the evidence of beneficial effects of regular physical activity on CV health and prevention of CVD risk, including BP, lipid levels and glucose tolerance [50] needs to be highlighted.

## Figures and Tables

**Figure 1 jcm-09-00655-f001:**
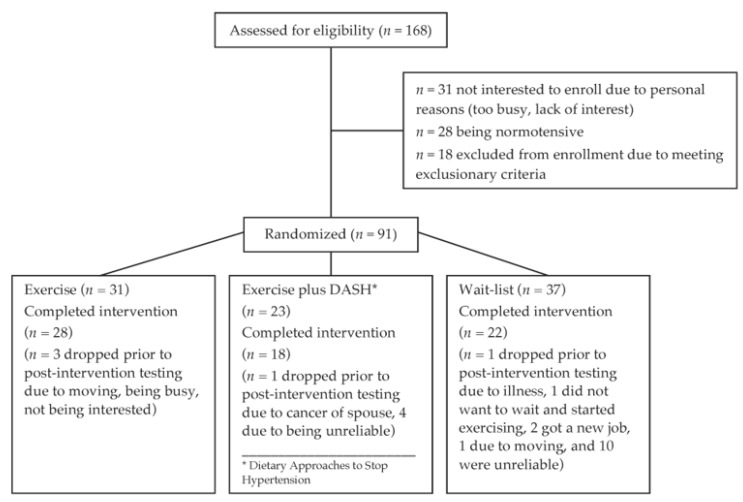
CONSORT flow diagram for participants.

**Table 1 jcm-09-00655-t001:** First step ^a^ and final ^b^ multivariate regression analyses between inflammatory biomarkers and intervention elements (predictors) in the studied pre- and hypertensive participants.

Predictor	Δ ^c^ Outcome ^d^	β	*p* ^e^
SBP ^e^	Δ PAI-1 (*n* = 62)	^a^ 0.415 ^b^ 0.427	0.003 0.001
Δ Fitness (VO_2peak_)	Δ PAI-1 (*n* = 62)	^a^ −0.369 ^b^ −0.392	0.004 0.006
Δ Caloric intake	Δ sICAM-3 (*n* = 63)	^a^ −0.363 ^b^ −0.332	0.014 0.026
	Δ CRP (*n* = 63)	^a^ 0.317 ^b^ 0.342	0.025 0.018
Δ Cardio training ^f^	Δ sICAM-3 (*n* = 63)	^a^ 0.314 ^b^ 0.301	0.037 0.046
	Δ IL-6 (*n* = 66)	^a^ 0.403 ^b^ 0.415	0.005 0.004
	Δ TNF-α (*n* = 68)	^a^ −0.337 ^b^ −0.357	0.018 0.017

^a^ The first step regression model included predictor, age, gender, and the difference from pre to post intervention biomarker level, and systolic blood pressure (except for systolic blood pressure as predictor). ^b^ The final model included the first step model and additionally the other predictors, and the difference from pre to post intervention body mass index level. ^c^ Δ, value difference from pre to post intervention level: We calculated the difference from pre to post intervention levels by detracting the value before the intervention from the value after the intervention. ^d^ PAI, plasminogen activator inhibitor; sICAM, soluble intercellular adhesion molecule; CRP, C-reactive protein; IL, interleukin; TNF-α, tumor necrosis factor-alpha. ^e^ SBP, systolic blood pressure. ^f^ Cardio training, moderate aerobic cardio training: total minutes during 12-week-intervention. ^e^
*p* < 0.05.

**Table 2 jcm-09-00655-t002:** Demographic, metabolic and health behavior characteristics of the study participants before the intervention.

Variable	*n* ^a^	Mean ± SEM and Range in Parentheses or Percentage Value
Prehypertension/Hypertension	68	44.1%/55.9%
Systolic blood pressure, mmHg	68	141 ± 1 (124–174)
Diastolic blood pressure, mmHg	68	84 ± 1 (69–102)
Heart rate, beats/min	68	75 ± 1 (53.0–99)
Age, years	68	45 ± 1 (25–60)
Gender, male/female	68	45%/55%
Body mass index, kg/m^2^	67	30.8 ± 0.5 (22.4–38.8)
Glucose, mg/dL	64	89 ± 1 (59–120)
Total cholesterol/HDL-cholesterol ratio	63	4.8 ± 0.2 (0.5–10.8)
Smoking	68	13%
Concomitant medication	68	0%
Leisure time exercise questionnaire	65	11.7 ± 1.2 (0.0–35.0)
VO_2peak_ ^b^ (mL/kg/min)	68	27.9 ± 0.9 (15.0–54.0)
Inflammatory measures before the intervention	*n* ^c^	
C-reactive protein, mg/L	63	2.8 (1.1–7.1)
Interleukin-6, pg/mL	66	2.7 (1.2–4.3)
Tumor necrosis factor-α, pg/mL	68	2.7 (1.2–4.1)
Plasminogen activator inhibitor-1, ng/mL	62	41.1 (21.4–61.2)
Soluble intercellular adhesion molecule-1, ng/mL	65	306.8 (239.3–374.1)
Soluble intercellular adhesion molecule-3, ng/mL	63	2.6 (2.2–3.2)

Data are given as mean ± standard error of the mean (SEM) (range) or percentage values. Inflammatory measures are given as medians (interquartile range). ^a^ n, number of subjects included. ^b^ VO_2peak_, peak oxygen consumption. ^c^ n, number of subjects who provided biomarker’s measurement before and after the intervention.

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
