# Peer review of "Effects of Caloric Intake and Aerobic Activity in Individuals with Prehypertension and Hypertension on Levels of Inflammatory, Adhesion and Prothrombotic Biomarkers—Secondary Analysis of a Randomized Controlled Trial"

_jcm, 2020, doi:10.3390/jcm9030655_

Round 1
Reviewer 1 Report
The paper from Wagner et al. describes a 12-week intervention to investigate the effects of cardio training, accompanied by caloric reduction or not, in prehypertensive and hypertensive adults. The low number of final participants and low compliance reduced the significance of the study, although the authors managed nonetheless to obtain interesting results. In fact, for the first time it was shown that adhesion molecules can be impacted by aerobic training, and diet. Further investigation is needed, though the study is of great interest for scientists in this research field.
Minor comments:
Line 48: “U.S.” Line 218: “significant” instead of “significance” Table 1 – first line: since it should be a percentage, add “%” like for the other percentages
Other comments:
The reviewer would add the number of males and females and prehypertensive/hypertensive in each group in the experimental design, to better describe it The reviewer suggest the calculation of ASCA to show the impact of factors such as sex on the results, to also better explain the necessity of the models adjustments. Were dietary records kept? If yes, this information needs to be added in the Methods section. How was intensity of exercise measured? Was it simply by the written logs, daily steps and reports from the PT? Was heart rate measured during exercise? Maybe more specifics on this are needed. Moreover, were all the pedometers the same? A great variance in reading can be experience from different kinds and brands, also more information on this might be useful.
Author Response
The paper from Wagner et al. describes a 12-week intervention to investigate the effects of cardio training, accompanied by caloric reduction or not, in prehypertensive and hypertensive adults. The low number of final participants and low compliance reduced the significance of the study, although the authors managed nonetheless to obtain interesting results. In fact, for the first time it was shown that adhesion molecules can be impacted by aerobic training, and diet. Further investigation is needed, though the study is of great interest for scientists in this research field.
Minor comments:
Line 48: “U.S.” Line 218: “significant” instead of “significance” Table 1 – first line: since it should be a percentage, add “%” like for the other percentages.
Response: Done as suggested:
- Line 48: “U.S.“
- Line 208 (former line 218): “significant“
- Table 1, first line: “44.1% / 55.9%“
Other comments:
The reviewer would add the number of males and females and prehypertensive/hypertensive in each group in the experimental design, to better describe it The reviewer suggest the calculation of ASCA to show the impact of factors such as sex on the results, to also better
explain the necessity of the models adjustments.
Response: We have searched online and have no idea what ASCA refers to. May we ask the Reviewer for clarification on that comment?
Were dietary records kept? If yes, this information needs to be added in the Methods section.
Response: Yes, dietry records were kept. We added this information to the METHODS section, lines 177-179:
„All subjects recorded their food (caloric intake) and provided the information to the study investigators.“
How was intensity of exercise measured? Was it simply by the written logs, daily steps and reports from the PT? Was heart rate measured during exercise? Maybe more specifics on this are needed. Moreover, were all the pedometers the same? A great variance in reading can be experience from different kinds and brands, also more information on this might be useful.
Response: We added the information required and clarified, METHODS section (lines 144-164: ”Since all participants were relatively sedentary at baseline, the … Exercise intensity was individually adapted by the assessed heart rate (before and during exercise). Participants met with a certified personal trainer (study personnel) at the local YMCA 2 days a week for the entire 12-week intervention period. They shared with the trainer the fitness information. Inperson
sessions with a trainer were scheduled, as they are more likely to be completed
than entirely home-based exercise and may increase the motivation for completion of the three sessions per week of exercise without a personal trainer. The sessions with the trainer included training in proper warm up, exercise and cool down and stretching. The participant was asked to do physical activity on his or her own (either in their home, neighborhood or at the YMCA) for three additional days per week. In addition to meeting a goal for the number
of minutes of physical activity, participants were asked to increase incidental physical activity. To monitor their progress, participants were given a pedometer and asked to work up to an ultimate goal of an average of 10,000+ steps per day. We used the same pedometer (Omron®) throughout the study to warrant an adequate comparison. The inclusion of the pedometer and step goal helped to insure that participants were not compensating for increased activity during their sessions with the personal trainer by being more sedentary throughout the rest of the day. The achievement of these goals ensured an increase in energy expenditure, which contributed to a negative energy balance. Subjects completed a log of their daily pedometer readings and presented them to the trainer each week.”

Reviewer 2 Report
Comments to the authors
The authors conducted a secondary analysis of a randomized controlled trial in which the effects of caloric restriction and aerobic activity in persons with prehypertension and hypertension were investigated. Outcome variables of this trial were biomarkers of inflammation, adhesion and pro-thrombosis. However, subjects in this trial apparently did not comply very well with their group allocation. This led the authors to pool all three groups for a secondary analysis. This is an interesting approach to a well-known problem in RCTs.
I would be good to have some information on how adherence to the dietary intervention was obtained. For the exercise intervention, this is explained well in the manuscript.
The secondary multivariate analysis revealed a number of associations between the predictors (blood pressure, fitness, caloric intake, and training) and the outcome biomarkers. There is no mention of a test for multiple comparisons or an explanation for not using one.
Otherwise, the manuscript is concise and well written. Length and discussion are appropriate. References should be more recent, if possible.
Major comments
The original RCT had largely varying group sizes which made me wonder why randomization was performed in such a way. In line 279, authors say that their earlier analyses are referenced in Ref 45 and 46. However, the reference list only contains 38 references. Did author do any test for multiple comparisons? If not, reasons should be stated. Table 2: There should be either asterisks or actual p-values in the last column, not both. How was caloric intake measured? The DASH diet has its own effects on hypertension, BMI and so on. It should be mentioned that this was not just a caloric reduction but also a profound dietary change. Were there food records from which nutrient intake could be obtained? Was adhesion to the DASH diet measured? If these things are mentioned in the publication of the RCT, please indicate this in the methods section. Otherwise please explain here. The most recent cited publication is from 2007. Are there really no recent papers to cite?
Minor comment
There are several measures that are inappropriately given with a decimal, such as age, heart rate, blood pressure, and glucose (abstract, results, table 1). The decimal does not add any relevant information and makes the content harder to grasp for the reader. Please remove. The first sentence of the introduction sounds odd. Please consider to use “to treat” and “drugs”. Line 89 might sound better “…from a clinical trial on the effects of…” Please correct the sentence in line 218. The abbreviation LFA-1 should be explained (line 317).
Author Response
Comments to the authors
The authors conducted a secondary analysis of a randomized controlled trial in which the effects of caloric restriction and aerobic activity in persons with prehypertension and hypertension were investigated. Outcome variables of this trial were biomarkers of inflammation, adhesion and pro-thrombosis. However, subjects in this trial apparently did not comply very well with their group allocation. This led the authors to pool all three groups for a secondary analysis. This is an interesting approach to a well-known problem in RCTs.
I would be good to have some information on how adherence to the dietary intervention was obtained. For the exercise intervention, this is explained well in the manuscript.
Response: We added this information to the METHODS section, lines 174-180:
“Thus, participants received intensive intervention approaches known to produce behavior change. Subjects in the experimental group met with registered dietitians and/or certified exercise trainers to establish initial dietary and physical activity goals. Regular meetings in person and by phone continued for the entire 12-week intervention. All subjects recorded their food (caloric intake in kilojoule) and provided the information to the study investigators. Diet adherence was assessed with three dayly dietary recalls administered by the study dietician.“
The secondary multivariate analysis revealed a number of associations between the predictors (blood pressure, fitness, caloric intake, and training) and the outcome biomarkers. There is no mention of a test for multiple comparisons or an explanation for not using one.
Response: Thank you for this comment. We clarified as recommended, lines 208-214:
“We did not adjust p-values for multiple comparisons, as the main primary outcome was CRP; however, as we examined the potential interaction among inflammatory, adhesion and prothrombic markers, we defined one primary outcome marker in each of these three biomarker domains playing different roles in CVD: CRP for inflammation, sICAM-1 for adhesion molecules, and PAI-1 for prothrombotic factors. All other outcomes (IL-6, TNF, sICAM-3) were secondary (Il-6 and TNF) or exploratory (sICAM-3, as no study so far has
been investigating sICAM-3).“
Otherwise, the manuscript is concise and well written. Length and discussion are appropriate. References should be more recent, if possible.
Response: We added several more recent references as suggested. However, we also kept older references, which we felt are important to present and discuss this hitherto underexplored topic the most comprehensibly.
References added:
“5. de Faria, A.P.; Ritter, A.M.; Sabbatini, A.R.; Corrêa, N.B.; Brunelli, V.; Modolo, R.; Moreno, H. Deregulation of Soluble Adhesion Molecules in Resistant Hypertension and Its Role in Cardiovascular Remodeling. Circ J. 2016, 80, 1196-201; DOI:10.1253/circj.CJ-16- 0058. Epub 2016 Apr 13.
8. Raggi, P.; Genest, J; Giles, J.T.; Rayner, K.J.; Dwivedi, G.; Beanlands, R.S.; Gupta, M. Role of Inflammation in the Pathogenesis of Atherosclerosis and Therapeutic Interventions. Atherosclerosis. 2018, 276, 98-108; DOI:10.1016/j.atherosclerosis.2018.07.014. Epub 2018 Jul 25.
11. Mirhafez, S.R.; Mohebati, M.; Feiz Disfani, M.; Saberi Karimian, M.; Ebrahimi, M.; Avan, A.; Eslami, S.; Pasdar, A.; Rooki, H.; Esmaeili, H.; et al. An Imbalance in Serum Concentrations of Inflammatory and Anti-inflammatory Cytokines in Hypertension. J. Am. Soc. Hypertens. 2014, 8, 614-623; DOI:10.1016/j.jash.2014.05.007. Epub 2014 May 21.
13. Horvei, L.D.; Grimnes, G.; Hindberg, K.; Mathiesen, E.B.; Njølstad, I.; Wilsgaard, T.; Brox, J.; Braekkan, S.K.; Hansen, J.B. C-reactive Protein, Obesity, and the Risk of Arterial and Venous Thrombosis. J. Thromb. Haemost. 2016, 14, 1561-1571; DOI:10.1111/jth.13369. Epub 2016 Jun 22.
36. Edwards, K.M.; Wilson, K.L.; Sadja, J.; Ziegler, M.G.; Mills, P.J. Effects on Blood Pressure and Autonomic Nervous System Function of a 12-week Exercise or Exercise Plus DASH-diet Intervention in Individuals With Elevated Blood Pressure. Acta Physiol. (Oxf). 2011, 203, 343-350; DOI:10.1111/j.1748-1716.2011.02329.x. Epub 2011 Jul 1.
37. Sadja, J.; Tomfohr, L.; Jiménez, J.A.; Edwards, K.M.; Rock, C.L.; Calfas, K.; Mills, P.J. Higher Physical Fatigue Predicts Adherence to a 12-week Exercise Intervention in Women With Elevated Blood Pressure. Health Psychol. 2012, 31, 156-163; DOI: 10.1037/a0025785. Epub 2011 Oct 10.“
Major comments
The original RCT had largely varying group sizes which made me wonder why randomizationwas performed in such a way.
Response: The study statistician determined the randomization sequence before the beginning of the study.
In line 279, authors say that their earlier analyses are referenced in Ref 45 and 46. However, the reference list only contains 38 references.
Response: We added the two missing references:
“36. Edwards, K.M.; Wilson, K.L.; Sadja, J.; Ziegler, M.G.; Mills, P.J. Effects on Blood Pressure and Autonomic Nervous System Function of a 12-week Exercise or Exercise Plus DASH-diet Intervention in Individuals With Elevated Blood Pressure. Acta Physiol. (Oxf). 2011, 203, 343-350; DOI:10.1111/j.1748-1716.2011.02329.x. Epub 2011 Jul 1.
37. Sadja, J.; Tomfohr, L.; Jiménez, J.A.; Edwards, K.M.; Rock, C.L.; Calfas, K.; Mills, P.J. Higher Physical Fatigue Predicts Adherence to a 12-week Exercise Intervention in Women With Elevated Blood Pressure. Health Psychol. 2012, 31, 156-163; DOI: 10.1037/a0025785. Epub 2011 Oct 10.“
Did author do any test for multiple comparisons? If not, reasons should be stated.
Response: We explained as recommended, lines 208-214:
“We did not adjust p-values for multiple comparisons, as the main primary outcome was CRP; however, as we examined the potential interaction among inflammatory, adhesion and prothrombic markers, we defined one primary outcome marker in each of these three biomarker domains playing different roles in CVD: CRP for inflammation, sICAM-1 for adhesion molecules, and PAI-1 for prothrombotic factors. All other outcomes (IL-6, TNF, sICAM-3) were secondary (Il-6 and TNF) or exploratory (sICAM-3, as no study so far has
been investigating sICAM-3).“
Table 2: There should be either asterisks or actual p-values in the last column, not both.
Response: Done as suggested.
How was caloric intake measured? The DASH diet has its own effects on hypertension, BMI and so on. It should be mentioned that this was not just a caloric reduction but also a profound dietary change. Were there food records from which nutrient intake could be obtained? Was adhesion to the DASH diet measured? If these things are mentioned in the publication of the RCT, please indicate this in the methods section. Otherwise please explain here.
Response: Yes, the Reviewer is right, the ulitmate goal was a profound dietary change. The caloric intake was measured in kilojoule, and nutrient intake could be obtained from food records. Diet adherence was assessed with three dayly dietary recalls administered by the study dietician. We added this information to the METHODS section, lines 174-180:
“Thus, participants received intensive intervention approaches known to produce behavior change. Subjects in the experimental group met with registered dietitians and/or certified exercise trainers to establish initial dietary and physical activity goals. Regular meetings in person and by phone continued for the entire 12-week intervention period. All subjects recorded their food (and caloric intake in kilojoule) and provided the information to the study investigators. Diet adherence was assessed with three dayly dietary recalls administered by the study dietician.“
The most recent cited publication is from 2007. Are there really no recent papers to cite?
Response: We added several more recent references as suggested. However, we also kept older references, which we felt are important to present and discuss this hitherto underexplored topic the most comprehensibly.
References added:
“5. de Faria, A.P.; Ritter, A.M.; Sabbatini, A.R.; Corrêa, N.B.; Brunelli, V.; Modolo, R.; Moreno, H. Deregulation of Soluble Adhesion Molecules in Resistant Hypertension and Its Role in Cardiovascular Remodeling. Circ J. 2016, 80, 1196-201; DOI:10.1253/circj.CJ-16- 0058. Epub 2016 Apr 13.
8. Raggi, P.; Genest, J; Giles, J.T.; Rayner, K.J.; Dwivedi, G.; Beanlands, R.S.; Gupta, M. Role of Inflammation in the Pathogenesis of Atherosclerosis and Therapeutic Interventions. Atherosclerosis. 2018, 276, 98-108; DOI:10.1016/j.atherosclerosis.2018.07.014. Epub 2018 Jul 25.
11. Mirhafez, S.R.; Mohebati, M.; Feiz Disfani, M.; Saberi Karimian, M.; Ebrahimi, M.; Avan, A.; Eslami, S.; Pasdar, A.; Rooki, H.; Esmaeili, H.; et al. An Imbalance in Serum Concentrations of Inflammatory and Anti-inflammatory Cytokines in Hypertension. J. Am. Soc. Hypertens. 2014, 8, 614-623; DOI:10.1016/j.jash.2014.05.007. Epub 2014 May 21.
13. Horvei, L.D.; Grimnes, G.; Hindberg, K.; Mathiesen, E.B.; Njølstad, I.; Wilsgaard, T.; Brox, J.; Braekkan, S.K.; Hansen, J.B. C-reactive Protein, Obesity, and the Risk of Arterial and Venous Thrombosis. J. Thromb. Haemost. 2016, 14, 1561-1571; DOI:10.1111/jth.13369. Epub 2016 Jun 22.
36. Edwards, K.M.; Wilson, K.L.; Sadja, J.; Ziegler, M.G.; Mills, P.J. Effects on Blood Pressure and Autonomic Nervous System Function of a 12-week Exercise or Exercise Plus DASH-diet Intervention in Individuals With Elevated Blood Pressure. Acta Physiol. (Oxf). 2011, 203, 343-350; DOI:10.1111/j.1748-1716.2011.02329.x. Epub 2011 Jul 1.
37. Sadja, J.; Tomfohr, L.; Jiménez, J.A.; Edwards, K.M.; Rock, C.L.; Calfas, K.; Mills, P.J. Higher Physical Fatigue Predicts Adherence to a 12-week Exercise Intervention in Women With Elevated Blood Pressure. Health Psychol. 2012, 31, 156-163; DOI: 10.1037/a0025785. Epub 2011 Oct 10.“
There are several measures that are inappropriately given with a decimal, such as age, heart rate, blood pressure, and glucose (abstract, results, table 1). The decimal does not add any relevant information and makes the content harder to grasp for the reader. Please remove.
Response: Done as suggested.
The first sentence of the introduction sounds odd. Please consider to use “to treat” and “drugs”. Line 89 might sound better “…from a clinical trial on the effects of…” Please correct the sentence in line 218. The abbreviation LFA-1 should be explained (line 317).
Response: Done as suggested.

Reviewer 3 Report
In this manuscript (ID#: jcm-728314), titled “Effects of Caloric Intake and Aerobic Activity in Individuals with Prehypertension and Hypertension on Levels of Inflammatory, Adhesion and Prothrombotic Biomarkers – Secondary Analysis of a Randomized Controlled Trial”, authors, Wagner et al, studied the effect of DASH diet and/or exercises on inflammatory, adhesion, and prothrombotic biomarkers in pre- or hypertensive patients. They conclude that regular exercise and caloric reduction may affect differentially the ability to induce the effects of cytokines on endothelial adhesion. This study area is very important. However, there are several major concerns, which are listed in the following paragraphs:
Figure 1 occupies too much space in the manuscript and should be condensed. This study lack normal control. Thus, we do not know whether the inflammatory, adhesion, and prothrombotic factors are elevated in hypertension and prehypertension as compared with normal individuals. Due to this weakness in experimental design, we do not know whether the changes induced by diet and exercises in these factors are beneficial or detrimental.
In Figure 1, I was confused by the observation grouping: group 1 exercise, group 2 exercise plus DASH, and group 3 Wait-list. What is the wait-list group? Another important group, DASH alone, is missing. The experimental design is so poor to examine the objective of this study. The table 1 should be spitted to several small tables, including 1) general information (e.g. age, smoking, …) in each group (control, prehypertension, and hypertension); 2) the blood pressure, heart rate, plasma Glucose, plasma lipids in each group (control, prehypertension, and hypertension) before and after treatments (exercises, exercises+DASH, and DASH); 3) the levels of inflammatory, adhesion, and prothrombotic factors in each group (control, prehypertension, and hypertension) before and after treatments (exercises, exercises+DASH, and DASH). In this manuscript, there is no clear conclusion reached: decrease or increase? Beneficial or detrimental? There is no explanation provided as well, in order to figure out the possible mechanisms involved.
Author Response
In this manuscript (ID#: jcm-728314), titled “Effects of Caloric Intake and Aerobic Activity in Individuals with Prehypertension and Hypertension on Levels of Inflammatory, Adhesion and Prothrombotic Biomarkers – Secondary Analysis of a Randomized Controlled Trial”, authors, Wagner et al, studied the effect of DASH diet and/or exercises on inflammatory, adhesion, and prothrombotic biomarkers in pre- or hypertensive patients. They conclude that regular exercise and caloric reduction may affect differentially the ability to induce the effects of cytokines on endothelial adhesion. This study area is very important. However, there are
several major concerns, which are listed in the following paragraphs:
Figure 1 occupies too much space in the manuscript and should be condensed.
Response: Yes, the Reviewer is right. Done as suggested (please see line 113).
This study lack normal control. Thus, we do not know whether the inflammatory, adhesion, and prothrombotic factors are elevated in hypertension and prehypertension as compared with normal individuals. Due to this weakness in experimental design, we do not know whether the changes induced by diet and exercises in these factors are beneficial or detrimental.
Response: Thank you for this comment. We now mention as a limitation of the study in the DISCUSSION, lines 327-331:
“... a weakness of the experimental design of this study is the lack of healthy controls. Thus, we do not know whether the inflammatory, adhesion, and prothrombotic factors are elevated in hypertension and prehypertension as compared with healthy individuals, and whether the changes induced by diet and exercise in these factors are beneficial or detrimental. Second,...“
In Figure 1, I was confused by the observation grouping: group 1 exercise, group 2 exercise plus DASH, and group 3 Wait-list. What is the wait-list group? Another important group, DASH alone, is missing. The experimental design is so poor to examine the objective of this study. The table 1 should be spitted to several small tables, including 1) general information (e.g. age, smoking, …) in each group (control, prehypertension, and hypertension); 2) the blood pressure, heart rate, plasma Glucose, plasma lipids in each group (control, prehypertension, and hypertension) before and after treatments (exercises, exercises+DASH, and DASH); 3) the levels of inflammatory, adhesion, and prothrombotic factors in each group (control, prehypertension, and hypertension) before and after treatments (exercises, exercises+DASH, and DASH). In this manuscript, there is no clear conclusion reached: decrease or increase? Beneficial or detrimental? There is no explanation provided as well, in order to figure out the possible mechanisms involved.
Response: Thank you for these important comments, we addressed as follows:
First, we would like to clarify the intervention groups. In order to explain the „wait-list control group“ better, we added to lines 106-108:
“Participants assigned to the wait-list group were controls for at least 12 weeks until they were allocated to one of the intervention groups (exercise alone or exercise plus DASH group).“
This information is also stated in lines 101-103:
“That study was conducted in order to compare the effect of an exercise program versus a combined exercise and diet program...“
Further, the RCT study compared three groups which were a wait-list control group, an exercise alone group and an exercise+DASH group, but it lacked a DASH alone group, as shown in Figure 1 and in lines 193-195:
“According to the RCT protocol, n = 18 had been to the aerobic cardio training and caloric reduction intervention, n = 28 to the aerobic cardio training alone intervention and n = 22 to the wait-list control group.“
Second, based on the modification of exercise or diet behavior of participants regardless of their intervention group assignment, we decided for a statistical approach with a secondary analysis of data from all 68 participants combined. We mentioned this procedure in lines 195-199:
“But in reality, participants began modifying their exercise or diet behavior regardless of their intervention group assignment and modest group sizes limited statistical power. Therefore we decided for a statistical approach with a secondary analysis of data from all 68 participants combined that may better warrant to indicate the actual behavior and inflammatory processes and consider the intervention effect described respectively.“
Following this statistical approach we edited Table 1 for a clearer presentation of the study sample as suggested above. However, we are afraid that a splitting of the whole sample into groups as suggested by this Reviewer might confuse readers, as the data were not analyzed between the three intervention groups.
Third, we extended on a potential role of sICAM-3, i.e. our conclusion and explanation of possible mechanisms involved as suggested, in lines 306-326:
“To date, studies on the effects of human hypertension on adhesion molecules have been focusing on sICAM-1 levels, which are increased in hypertensive individuals [4,5]. Soluble ICAM-1 levels reflect ICAM-1 expression on activated endothelial cells [35] and are associated with the degree of atherosclerosis [36]. Although ICAM-1 and ICAM-3 share a similar immunoglobulin-like structure and amino acid identity of about 48%, with the greatest homology observed in domains 2 and 3 [37], their differential pattern of expression and cellular distribution suggest a different functional role [37]. ... Although sICAM-3 binds to LFA-1 with an affinity approximately 9 times weaker than sICAM-1, sICAM-1 and sICAM-3 compete with each other for binding to LFA-1 [37]. As we did not find changes in sICAM-1 levels post-interventions in this study, we speculate that the increase in sICAM-3 levels as a result of changes in exercise and/or diet behavior in this study may exhibit a “buffer” for the sICAM-1 function, potentially leading to downregulating vascular endothelial inflammation. It is not fully understood whether the increase of sICAM-3 by cardio training and caloric reduction in prehypertensive and hypertensive subjects reflects a cardio-protective mechanism. Given the lack of knowledge in the vascular function of sICAM-3 and its implications as a vascular inflammatory biomarker, a follow-up investigation is necessary to examine its vascular action and interactions separate of or in conjunction with other betterknown vascular endothelial markers such as sICAM-1.“

Round 2
Reviewer 3 Report
The revised manuscript has been improved. No additional concern.
Author Response
Responses to Reviewer 1 (Round 2)
ASCA is: ANOVA-simultaneous component analysis
Response: We thank the Reviewer for this thoughtful comment and clarification. However, after consulting with a biostatistician we have determined that ASCA is more appropriate for high dimensional data with a large array of outcome measures and time points. Thus, we have decided to maintain our current analytical approach. We are happy to adress this further if desired, however. We addressed this comment in the LIMITATIONS, lines 370-373:
“Fourth, our study did not explore the role of gender in the potentially differing effects of exercise and / or diet on inflammatory or prothrombotic markers, which was beside the primary aim of the study and limited by a limited sample size. Future studies should investigate the effect of gender not only on the intervention effects but also such behavior changes.”
